# Spark Plasma Sintering of Ceramics Based on Solid Solutions of $Na_{1+2x}Zr_{2−x}Co_x(PO_4)_3$ Phosphates: Thermal Expansion and Mechanical Properties Research

A. A. Aleksandrov [1], A. I. Orlova [1,*], D. O. Savinykh [1], M. S. Boldin [1], S. A. Khainakov [2], A. A. Murashov [1], A. A. Popov [1], G. V. Shcherbak [1], S. Garcia-Granda [2], A. V. Nokhrin [1,*], V. N. Chuvil'deev [1] and N. Yu. Tabachkova [3,4]

[1] Materials Science Department, Physical and Technical Research Institute, Lobachevsky State University of Nizhny Novgorod, 603022 Nizhny Novgorod, Russia
[2] Faculty of Chemistry, University of Oviedo, 33003 Oviedo, Spain
[3] Center Collective Use "Materials Science and Metallurgy", National University of Science and Technology "MISIS", 119991 Moscow, Russia
[4] Laboratory "FIANIT", Laser Materials and Technology Research Center, A.M. Prokhorov General Physics Institute of the Russian Academy of Sciences, 119991 Moscow, Russia
* Correspondence: albina.orlova@gmail.com (A.I.O.); nokhrin@nifti.unn.ru (A.V.N.)

**Abstract:** The structure, microstructure, coefficient of thermal expansion (CTE), and mechanical properties of $Na_{1+2x}Zr_{2−x}Co_x(PO_4)_3$ ceramics (x = 0, 0.1, 0.2, 0.3, 0.4, 0.5) were studied. $Na_{1+2x}Zr_{2−x}Co_x(PO_4)_3$ submicron powders with the $NaZr_2(PO_4)_3$ structure (NZP, kosnarite type) were obtained by the solid-phase method. The starting reagents ($NaNO_3$, $ZrOCl_2·8H_2O$, $NH_4H_2PO_4$, $CoCl_2·6H_2O$, ethanol) were mixed with the addition of ethyl alcohol. The resulting mixtures were annealed at 600 °C (20 h) and 700 °C (20 h). The obtained phosphates crystallized in the expected structure of the $NaZr_2(PO_4)_3$ type (trigonal system, space group R$\bar{3}$c). Thermal expansion of the powders was studied with high-temperature X-ray diffraction at temperatures ranging from 25 to 700 °C. CTEs were calculated, and their dependence on the cobalt content was analyzed. $Na_{1+2x}Zr_{2−x}Co_x(PO_4)_3$ ceramics with high relative density (93.67–99.70%) were obtained by Spark Plasma Sintering (SPS). Ceramics poor in cobalt (x = 0.1) were found to have a high relative density (98.87%) and a uniform fine-grained microstructure with a grain size of 0.5–1 μm. Bigger cobalt content leads to a smaller relative density of ceramics. During the sintering of ceramics with high cobalt content, anomalous grain growth was observed. The powder compaction rate was shown to be determined by creep and diffusion intensity in the $Na_{1+2x}Zr_{2−x}Co_x(PO_4)_3$ crystal lattice. SPS activation energy in ceramics increased as the cobalt content grew. The microhardness and fracture toughness of ceramics did not depend on their cobalt content.

**Keywords:** NZP; isomorphism; sintering; microstructure; X-ray diffraction; thermal expansion

## 1. Introduction

Inorganic compounds with the kosnarite-type $NaZr_2(PO_4)_3$ structure (NZP, NASICON) are considered to be promising materials in terms of the immobilization of highly active components of radioactive waste (high-level waste, HLW) [1–9]. Inorganic compounds with the $NaZr_2(PO_4)_3$ structure allow for an opportunity to implement an HLW isolation concept by means of the "atomic-level inclusion" of these compounds into crystal lattices of novel ceramic materials characterized by high structural stability when subjected to ultralong-term exposure to various extremal factors: radiation, hydrothermal, and thermal actions [10–22]. These findings seem to be highly relevant in tackling the utilization and safe storing of radiochemical industry wastes [1,2,5–7,9,12,23–36].

Low resistance to thermal shocks, as compared to construction ceramics, is one of the drawbacks of bulk mineral-like ceramic materials (ceramic matrices) [36,37]. It can lead

to microcracks that form under the high-speed heating or cooling of ceramic matrices (for example, in the case of high-speed ceramic matrices compaction methods). Formation of micro- and macrocracks can increase the free surface area and, as a consequence, intensify the HLW leaching rate from ceramics. In the future, it can lead to accelerated HLW leakage into ground waters and radiation pollution of the environment. To tackle this challenging issue, two-phase composites are being developed at present; the individual phases in which have different thermal expansion coefficients (CTE) [36,38–42]. A proper choice of phases with given CTEs and their ratio allows us to "control" internal stress fields that form compressive internal stress fields in ceramics [36,40] and, hence, increase their fracture toughness (crack resistance).

It should be noted that materials that are in contact with each other and have different CTEs are subjected to local strains under the influence of thermal stress, which ultimately leads to the failure of ceramic matrices. Therefore, it is very important to search for and create such materials whose CTEs can be adjusted in line with practical goals.

In our opinion, ceramics whose NZP structure has an isomorphism phenomenon have great prospects for providing high resistance to thermal shocks. In the future, the phenomenon of isomorphism may make it possible to create new pseudo-polyphase ceramics sintered from structures of the same type (for example, kosnarite-type) but with different CTEs.

Employing polymorphism, one can control CTEs and, in particular, reduce the anisotropy of CTE in ceramic structures [43–55]. CTE anisotropy reduction is a known method of bringing down internal stresses in ceramics, particularly under thermal shocks. The isomorphism phenomenon mentioned above can help to develop novel pseudo-polymorphous ceramics sintered from the structures of the same types with different CTEs (see, for example, refs. [37,38,56]).

The more diverse the number of potential cations or anions the structure can include, the wider the possibilities to regulate the properties of the material. $NaZr_2(PO_4)_3$ (NZP) can be one of these structures. Structural analogs of $NaZr_2(PO_4)_3$ (NZP) are promising compounds with controlled physicochemical properties. Compounds of the NZP group studied earlier show thermal, chemical, and radiation resistance [1–5,7,10–23,26,29–35,42,50] and low thermal expansion [15,57–66].

The NZP structure is a framework [1–3,8,11,33,35,46,63,64]. The framework is formed by zirconium octahedrons ($ZrO_6$) and phosphate tetrahedrons ($PO_4$) bound to each other with common oxygen atoms [1,3]. Blocks of two octahedrons and three tetrahedrons form a three-dimensional framework containing cavities, predominantly filled with large low-charged cations. In general, compounds with such a structure are described by the following crystallochemical formula: $(M1)^{VI}(M2)^{VIII}_3[L^{VI}_2(PO_4)_3]$, where M1 and M2 are positions in the framework cavities. L stands for the positions of zirconium, while VI and VIII are co-ordination numbers (CN) in these positions.

The NZP structural type is characterized by broad isomorphism and has a wide variability of the elemental composition [1,3,33,34,67,68]. The NZP-type structure in cavities of this structure contain cations such as Li, Na, K, Rb, Cs, Cu, and Ag; Mg, Ca, Sr, and Ba; Mn, Co, Ni, Cu, Zn, and Cd; Sc, Fe, Bi, Ce–Lu, Am, and Cm; Zr, Hf, Th, U, Np, and Pu may be placed. Furthermore, the frameworks of this structure can contain V, Nb, Sb, and Ta; Ti, Ge, Zr, Hf, U, Np, Pu, Mo, and Sn; Al, Sc, Cr, Fe, Ga, Y, and In; Gd, Tb, Dy, Er, and Yb; Mg; Na and K. The properties of the resulting compounds differ significantly depending on the specific isomorphic substitution. The list of papers on the isomorphism of NZP structures is very long (see, for example, refs. [1,2,67,68]).

The behavior of NZP compounds after heating depends on the nature of the cations included in their composition (their size, charge, and electronegativity); it is based on the contribution of mainly the weakest bonds. Therefore, the effect of cations located in extraframework M positions and the occupancy of these positions are more pronounced [44,45,49,50,52,68–77]. Many compounds of the NZP family have low CTEs. It

was found that the substitution of $Zr^{4+}$ for $Fe^{3+}$ cations [77] promotes a decrease in the absolute values of linear CTEs and anisotropy.

The sintering method is crucial in developing ceramic materials for HLW immobilization and other forward-looking applications in nuclear power engineering. One of the promising ways to obtain ceramic materials is Spark Plasma Sintering (SPS) [78]. Materials obtained this way have high relative density [2,10,36,40,41,77–87]. SPS of ceramics is carried out as follows: pulsed direct current of high power is passed through a specimen powder and the tooling under pressure in a vacuum or inert environment [78]. Ceramic materials obtained using this method have enhanced physical and mechanical properties, which make it possible to obtain various materials for different functional purposes. An opportunity to obtain ceramics with high heating rates (up to 2500 °C/min) and at reduced sintering temperatures is a fundamental advantage of SPS technology in nuclear power engineering [2,78,83–98]. Thus, it becomes possible to dramatically reduce dissociation rates of dangerous radwaste components from the surfaces of ceramic matrices during sintering [2]. In our opinion, SPS technology can be considered an efficient alternative to traditional hot sintering technologies or sintering pre-compacted powders, which are now used to obtain ceramic matrices [2,78].

It is worth mentioning that ceramic mineral-like materials have low diffusion coefficients and, hence, are difficult to sinter using conventional methods; sometimes, typical sintering times exceed several hours [2,82]. Therefore, fusible additives that allow us to reduce the sintering temperatures of mineral-like materials are often used to accelerate the sintering of ceramic materials [88,89]. SPS technology helps to obtain highly dense ceramic materials without extra additives, i.e., sintering activators [2,10,36,40,41,77,82,83,85,87,98].

At present, SPS technology is used extensively for various materials in nuclear power engineering, in particular, nuclear fuel cycle materials [85,92–95], inert fuel matrices [2,36,40,41,90,95,96], and ceramics for HLW immobilization [2,10,77,82,83,85,97]. The efficiency of SPS technology in terms of producing NZP ceramics was demonstrated in ref. [2]. SPS technology was found to allow a uniform fine-grained microstructure with high relative density and high hydrolytic resistance in NZP ceramics.

This study aims to design new NZP ceramic materials based on solid solutions of phosphates with a directed change in the chemical composition to control the isomorphism effect, to study the dependence of CTEs on specimen composition, and to study relevant mechanical properties.

## 2. Materials and Methods

Solid solutions of $Na_{1+2x}Zr_{2-x}Co_x(PO_4)_3$ composition (x = 0.1–0.5) make up the target of this research. Zirconium and cobalt have close ionic radii (r($Zr^{4+}$) = 0.072 nm; r($Co^{2+}$) = 0.074 nm); besides, cobalt adds color to materials, which can be useful in some designs.

Powder synthesis was carried out using the solid-phase method. The starting reagents were sodium nitrate $NaNO_3$ (98%), zirconium oxychloride $ZrOCl_2 \cdot 8H_2O$ (99%), ammonium dihydrogen phosphate $NH_4H_2PO_4$ (99%), cobalt chloride $CoCl_2 \cdot 6H_2O$ (99%), and ethanol (97%). Cobalt chloride was calcined at 350 °C. The mixture of reagents taken in stoichiometric ratios was thoroughly dispersed in an agate mortar with the addition of ethyl alcohol for 15 min. Then, these mixtures were annealed at 600 °C for 20 h to remove nitrates and volatile compounds. The resulting mixture was ground up with the addition of ethyl alcohol for 20 min and annealed at 700 °C for 20 h.

Ceramics were obtained with SPS using the Dr. Sinter model SPS-625 setup (SPS SYNTEX, Japan) from synthesized powders. The powders were placed in a graphite die with an inner diameter of 12 mm and heated by passing millisecond pulses of high-power direct electric current (up to 3 kA). The temperature was measured with a Chino IR-AH pyrometer focused on the surface of a graphite die. Sintering was carried out in a vacuum (6 Pa). The uniaxial stress applied was 70 MPa. Sintering involved a two-stage heating process: heating started at the rate of 100 °C/min and reached 600 °C, then heating rates lowered to 50 °C/min while sintering temperatures reached $T_s$ = 850–970 °C. There was no

holding time ($t_s$ = 0 s). Sintered ceramic specimens were cooled down using the Dr. Sinter model SPS-65. The effective shrinkage ($L_{eff}$) and shrinkage rate ($S_{eff}$) of the powders were monitored with a dilatometer built into the Dr. Sinter model SPS-625 setup. True shrinkage (L) and shrinkage rate (S) were calculated by subtracting the thermal expansion of the setup ($L_0$) from $L_{eff}$. The $L_0$(T) curve was determined during experiments that involved heating an empty die in the same modes as the ones used while sintering ceramic matrices.

X-ray diffraction (XRD) phase analysis was performed using a Shimadzu LabX XRD 6000 X-ray diffractometer with CuK$\alpha$ radiation, $\lambda$ = 1.54056 Å in the angular range 2$\theta$ = 10–50°. Crystal lattice parameters were calculated using GSAS-II. High-temperature recording of XRD spectra was carried out with the help of a Panalytical X'Pert Pro diffractometer using an Anton Paar HTK-1200N high-temperature chamber at temperatures ranging from 25 to 700 °C. CTE was determined based on the slope of dependence of the crystal lattice parameter on the measurement temperature. In the course of the experiment, the CTE values along the a ($\alpha_a$) and c ($\alpha_c$) axes, the CTE ($\alpha_{av}$) average value, and the bulk CTE ($\beta$) were determined. The average CTE was calculated using the following formula: $\alpha_{av} = (2\cdot\alpha_a + \alpha_c)/3$.

The density ($\rho$) of the ceramics obtained was measured by hydrostatic weighing in distilled water with a Sartorius CPA 225D balance. The density accuracy was $\pm0.001$ g/cm$^3$. The theoretical density of the ceramics ($\rho_{th}$) was calculated based on the results of X-ray studies.

The microstructure of powders and ceramics was studied with a JEOL JSM-6490 scanning electron microscope (SEM) with an Oxford Instruments INCA 350 EDS microanalyzer and Jeol® JEM-2100F Transmission Electron Microscope (TEM).

The microhardness (Hv) of the ceramics was measured using a Qness A60+ hardness tester. The load was 500 g. The minimum fracture toughness coefficient ($K_{IC}$) was calculated using the Palmquist method based on the length of the largest radial crack formed when the ceramic was indented with a Vickers pyramid. The average error in determining Hv and $K_{IC}$ was $\pm0.2$ GPa and $\pm0.2$ MPa$\cdot$m$^{1/2}$, respectively.

Prior to investigations of microstructure and mechanical properties, the sintered ceramic specimens were sequentially subjected to annealing in an air furnace (700 °C, 2 h), mechanical grinding, and polishing. Annealing and mechanical treatment of the surfaces were used to remove residual graphite and carbonized surface layers formed during the high-temperature interaction of ceramics with graphite [99–105]. Grinding and polishing were performed using a Struers® Secotom™ 10 grinding machine and a Buehler® Automet™ 250 polishing setup.

### 3. Results and Discussion

Synthesized specimens included polycrystalline powders ranging in color from light blue to violet, depending on the cobalt content. According to SEM and TEM data, the powders obtained were agglomerates of particles with a nonuniform size distribution (Figures 1 and 2). The agglomerates of 5–20 μm in size consisted of submicron particles of <0.1–0.2 μm. The size of powder particles did not depend on the cobalt content; powders containing cobalt had a nonhomogeneous granulometric composition: large submicron faceted particles of ~200–500 nm in size were observed on the background of nanoparticles of ~50 nm in size (Figure 2c,d). Synthesized nanopowders retained their crystal structure (Figure 2e,f). Some submicron particles consisted of rounded nanoparticles sintered together in the course of high-temperature annealing (700 °C, 20 h) during solid-phase synthesis (Figure 2g,h). A thin amorphous layer was visible on the surface of the nanoparticles (Figure 2i,j). According to XRD data (Figure 3), the solid solutions under study crystallized in a NaZr$_2$(PO$_4$)$_3$ structure (trigonal system, R$\overline{3}$c space group). No XRD peaks from auxiliary (impurity) phases were observed in the XRD curves (Figure 3). Peaks on diffraction patterns correspond to XRD peaks for the NaZr$_2$(PO$_4$)$_3$ compound (PDF #00-033-1312).

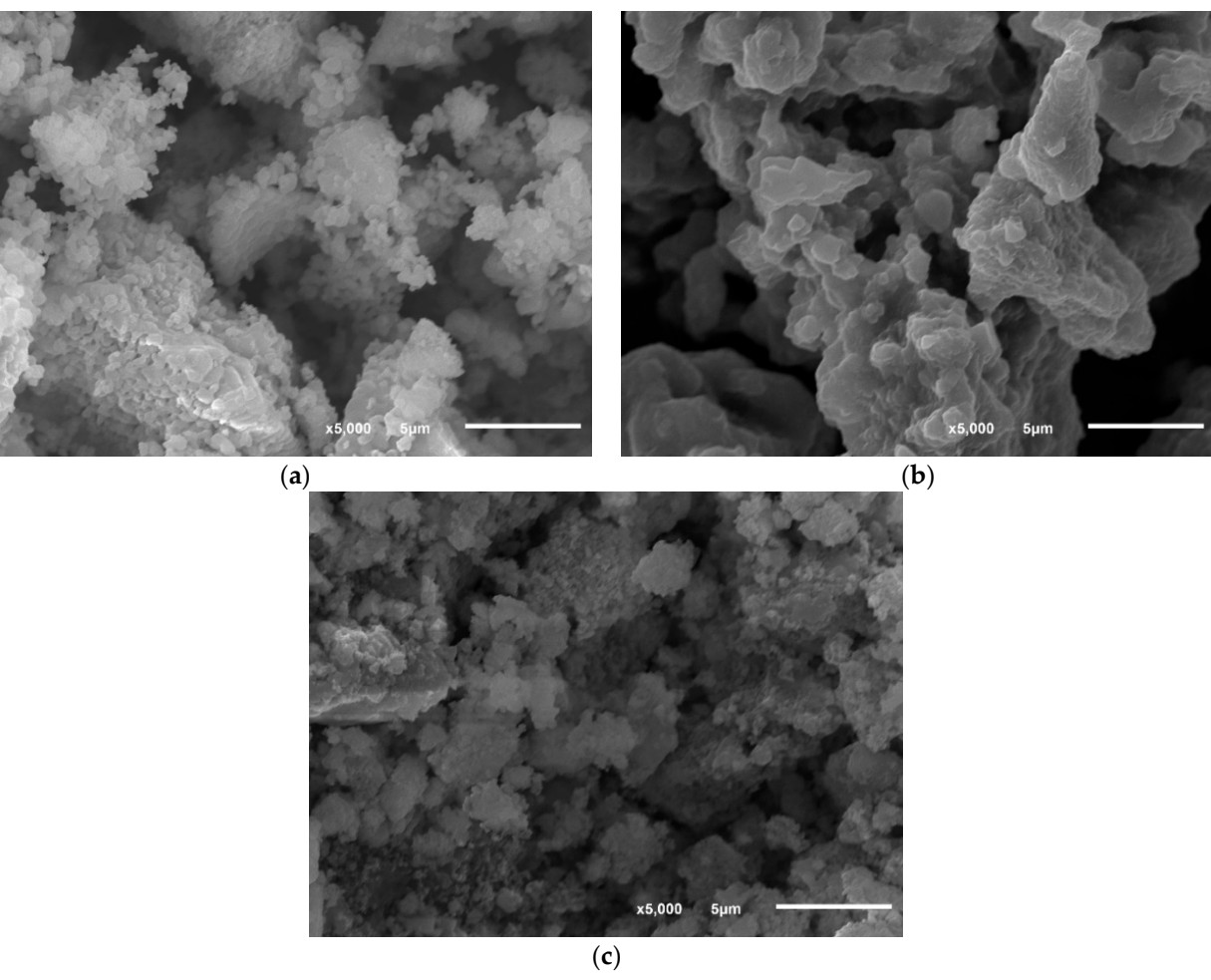

**Figure 1.** SEM micrographs of $Na_{1.2}Zr_{1.9}Co_{0.1}(PO_4)_3$ (**a**), $Na_{1.5}Zr_{1.75}Co_{0.25}(PO_4)_3$ (**b**), and $Na_2Zr_{1.5}Co_{0.5}(PO_4)_3$ (**c**) powders.

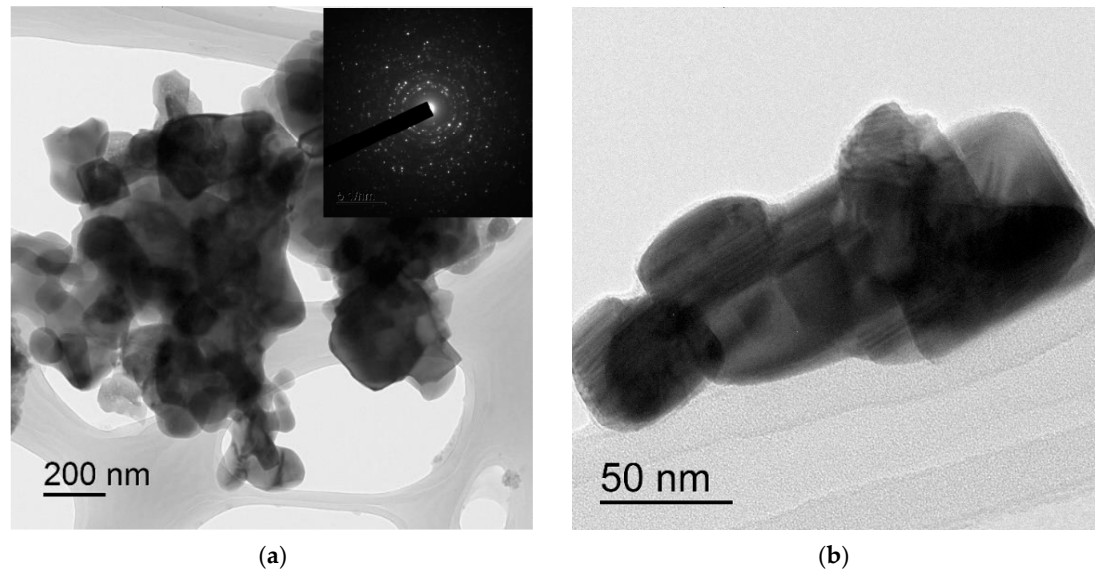

(**a**) (**b**)

**Figure 2.** *Cont.*

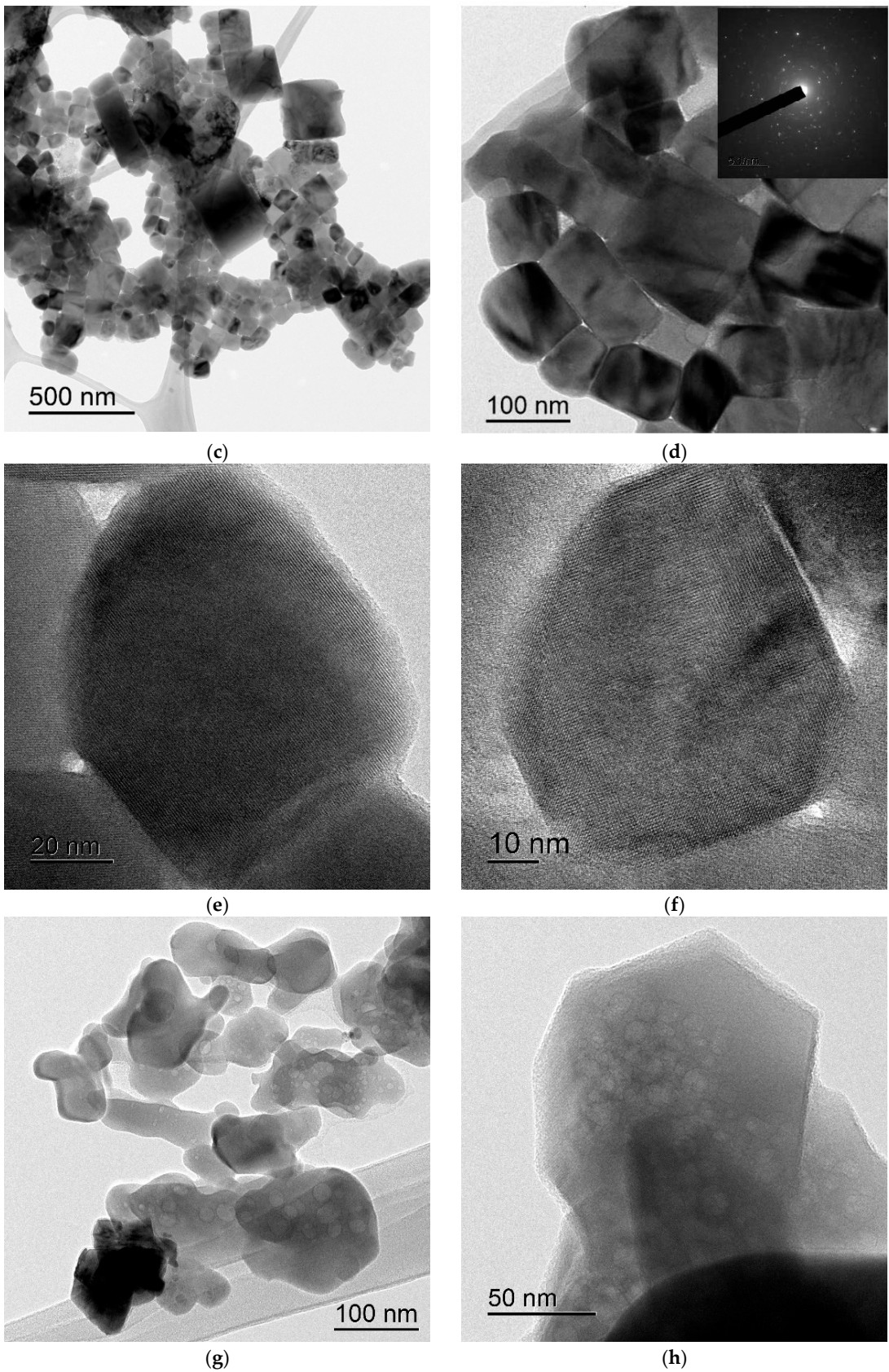

**Figure 2.** *Cont.*

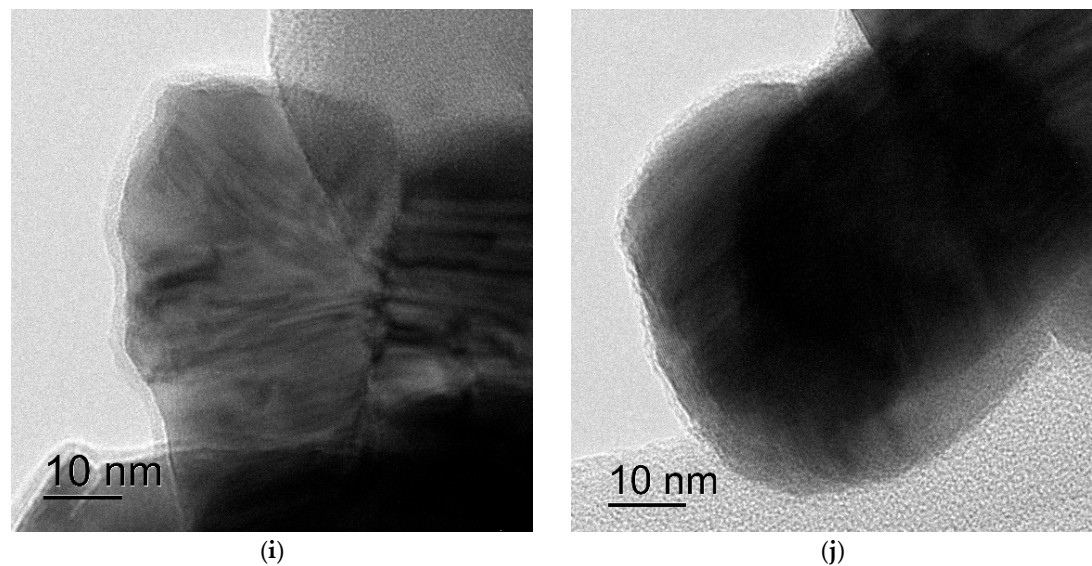

**Figure 2.** TEM micrographs of $Na_{1.2}Zr_{1.9}Co_{0.1}(PO_4)_3$ (**a,b**) and $Na_2Zr_{1.5}Co_{0.5}(PO_4)_3$ (**c–j**) powders.

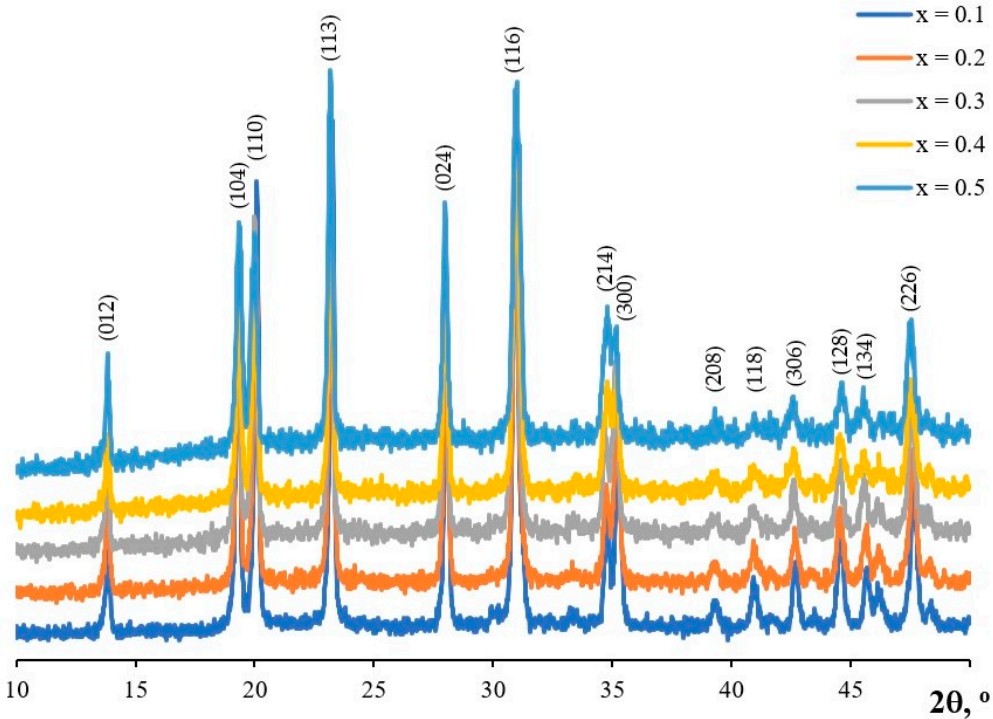

**Figure 3.** XRD data for powders $Na_{1+2x}Zr_{2-x}Co_x(PO_4)_3$. Diffraction angle indices are provided in accordance with the data for the $NaZr_2(PO_4)_3$ compound (PDF #00-033-1321).

The graphs of the dependence of unit cell parameters of $Na_{1+2x}Zr_{2-x}Co_x(PO_4)_3$ compounds on the composition are shown in Figure 4. It was found that with an increase in the cobalt content, there is a tendency towards an increase in the *a* parameter and a decrease in the *c* parameter. Lattice parameter values are provided in Table 1. The behavior of solid solution $Na_{1+2x}Zr_{2-x}Co_x(PO_4)_3$ specimens after heating at temperatures ranging from 25 to 700 °C was anticipated and typical of NZP compounds: a unit cell expands along the *c* axis and contracts along the *a* and *b* axes. This data is based on the results of high-temperature radiography (Figure 5).

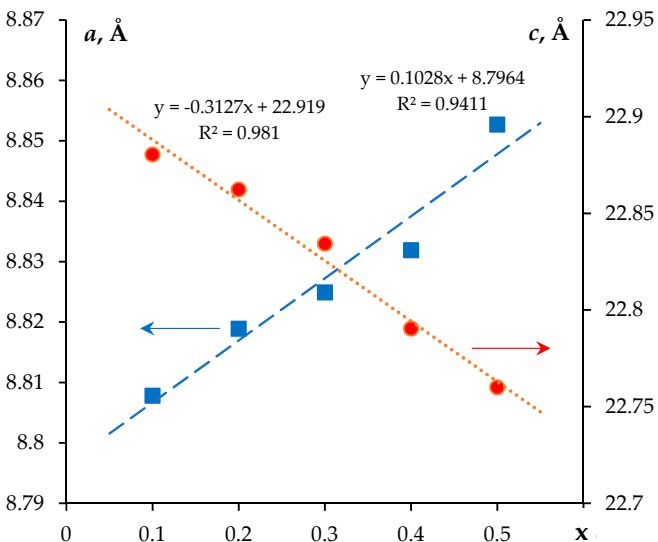

**Figure 4.** Dependences of crystal lattice parameters *a* and *c* on the composition of compounds $Na_{1+2x}Zr_{2-x}Co_x(PO_4)_3$. Analysis of the results is presented in Figure 3.

**Table 1.** Lattice parameters and thermal expansion parameters of $Na_{1+2x}Zr_{2-x}Co_x(PO_4)_3$ phosphates.

| x | 0.1 | 0.2 | 0.3 | 0.4 | 0.5 |
|---|---|---|---|---|---|
| a, Å | 8.807 (8) | 8.818 (9) | 8.824 (9) | 8.831 (9) | 8.852 (7) |
| c, Å | 22.880 (4) | 22.862 (3) | 22.834 (3) | 22.790 (4) | 22.760 (0) |
| $\alpha_a \times 10^6$, °C$^{-1}$ | −2.271 | −1.022 | −0.682 | −2.272 | −1.136 |
| $\alpha_c \times 10^6$, °C$^{-1}$ | 17.490 | 17.496 | 17.512 | 21.939 | 22.048 |
| $\alpha_{av} \times 10^6$, °C$^{-1}$ | 4.316 | 5.151 | 5.383 | 5.798 | 6.592 |
| $\beta \times 10^6$, °C$^{-1}$ | 14.253 | 14.742 | 16.109 | 18.577 | 17.894 |
| $\Delta\alpha \times 10^6$, °C$^{-1}$ | 19.761 | 18.518 | 18.194 | 24.211 | 23.184 |

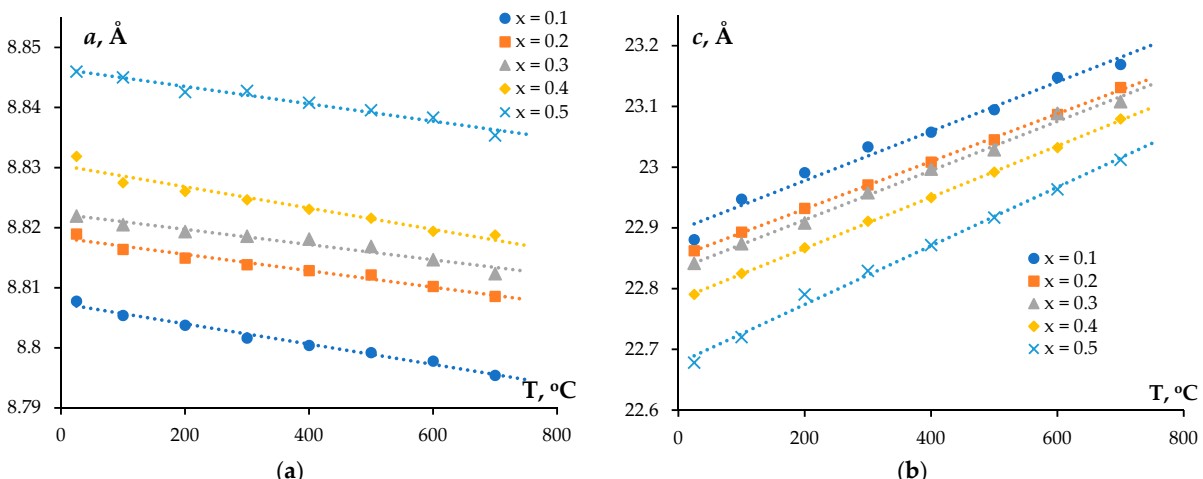

**Figure 5.** Temperature dependences of crystal lattice parameters *a* (**a**) and *c* (**b**) of $Na_{1+2x}Zr_{2-x}Co_x(PO_4)_3$ compounds.

The analysis offered in Table 1 and CTE values, axial ($\alpha_a$ and $\alpha_c$), average ($\alpha_{av}$), and bulk ($\beta$), show that an increase in the cobalt content leads to an increase in $\alpha_{av}$, which corresponds to conventional behavior of NZP materials (Figure 6). Cavity volume increases with an increase in the cobalt content; therefore, atoms in the crystal lattice vibrate in a larger space than in $NaZr_2(PO_4)_3$. It is impossible to establish dependences on composition changes for $\alpha_a$, $\alpha_c$, $\Delta\alpha$, and $\beta$.

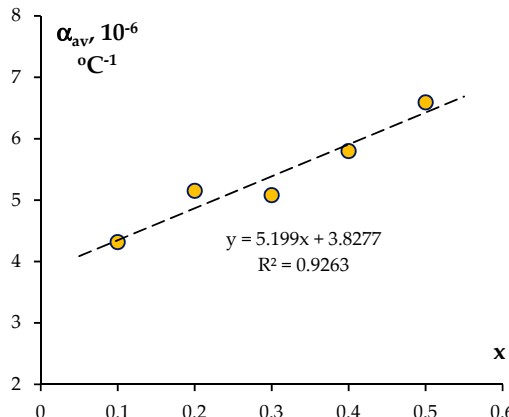

**Figure 6.** Dependences of the $\alpha_{av}$ parameters on the composition $Na_{1+2x}Zr_{2-x}Co_x(PO_4)_3$.

In our opinion, such an anomalous behavior of the material is due to the fact that cobalt atoms are located not only in the intraframework L position but also in the M2 position (see Appendix A). Unfortunately, at present, it is impossible to provide an unambiguous answer regarding the nature of the Co distribution between possible L and M2 positions. Single-crystal X-ray diffractometry is the most effective method to decipher the structure. To ensure effective application of this technique, it is necessary to use specimens of single crystals at least 30–100 μm in size. Unfortunately, currently, there is no chance to prepare a single crystal of the $Na_{1+2x}Zr_{2-x}Co_x(PO_4)_3$ compound using the solid-phase method or wet chemistry method.

Ceramics were sintered using SPS from prepared powders of $Na_{1+2x}Zr_{2-x}Co_x(PO_4)_3$ phosphates (x = 0.1, 0.25, 0.5). Sintering time did not exceed 1 min. The dependences of temperature (T), applied pressure (P), and vacuum pressure (Vac) on SPS time are shown in Figure 7. The amount of uniaxial applied pressure and vacuum level was almost constant during heating. There was no intensive decrease in the vacuum pressure, which allows us to conclude that the ceramic crystal structure is thermally stable during heating, and there is no intense dissociation of elements from the specimen surface.

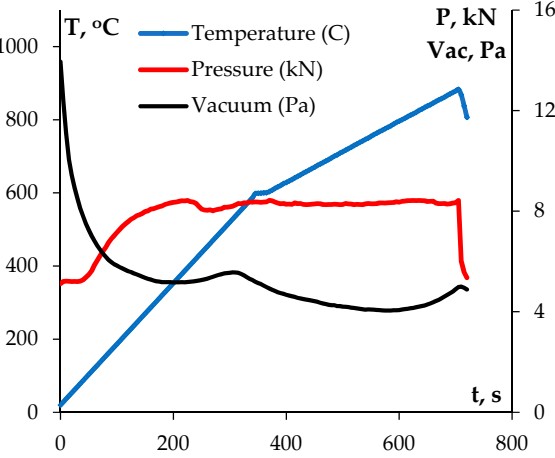

**Figure 7.** Dependences of temperature (T), applied pressure (P), and vacuum pressure (Vac) on SPS time for $Na_{1.2}Zr_{1.9}Co_{0.1}(PO_4)_3$ ceramics.

Ceramic specimens with a diameter of 12 mm were obtained without visible micro- and macro-cracks (Figure 8). Relative density values are presented in Table 2. Both relative and absolute density of sintered ceramic specimens decreased from 98.87% (3.151 g/cm³) to 93.30% (3.006 g/cm³), along with an increase in cobalt content. No large pores were found on the surface of the ceramic specimens.

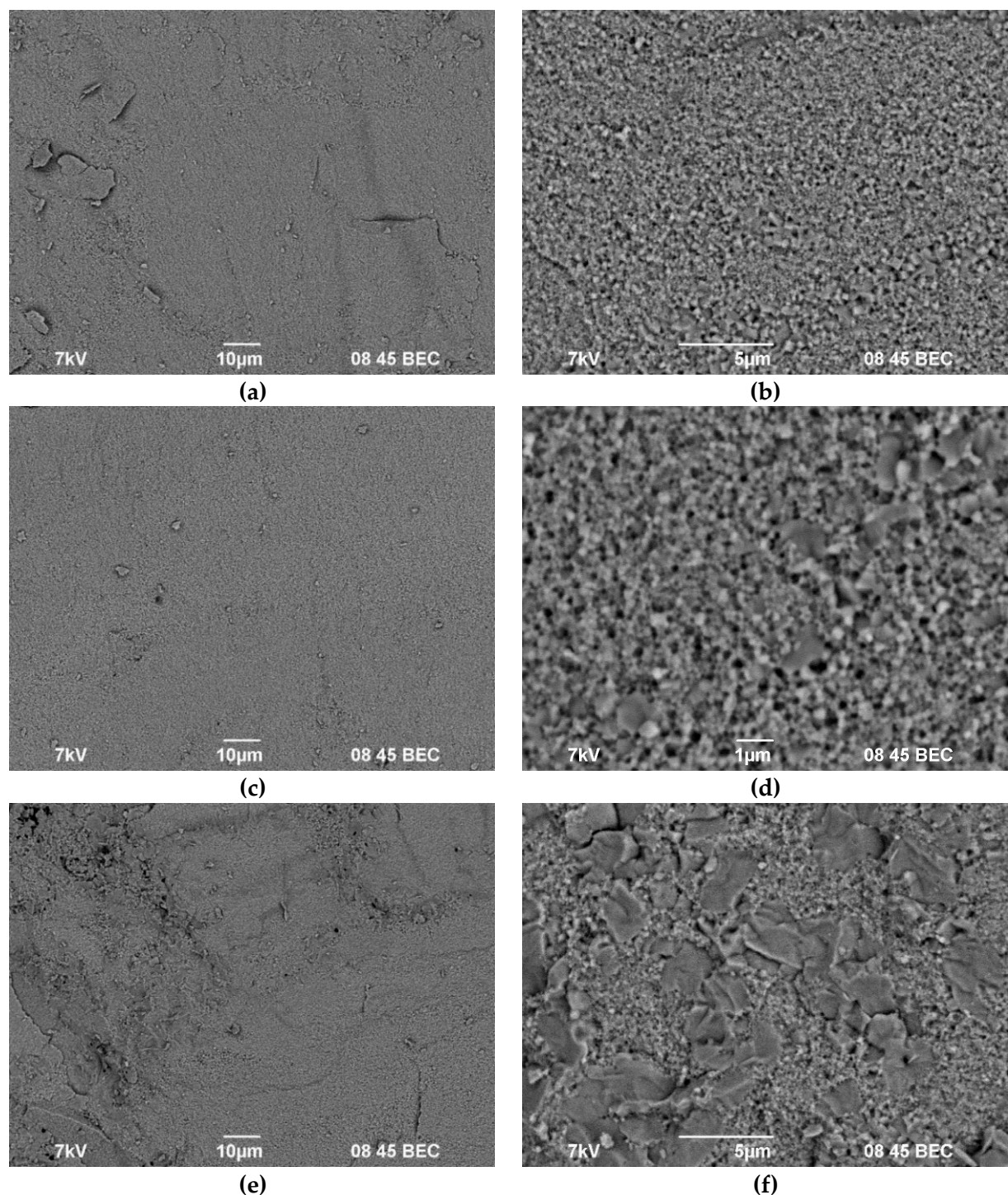

**Figure 8.** SEM images of macrostructure (**a**,**c**,**e**) and microstructure (**b**,**d**,**f**) of $Na_{1.2}Zr_{1.9}Co_{0.1}(PO_4)_3$ (**a**,**b**), $Na_{1.5}Zr_{1.75}Co_{0.25}(PO_4)_3$ (**c**,**d**), and $Na_2Zr_{1.5}Co_{0.5}(PO_4)_3$ (**e**,**f**) ceramics.

**Table 2.** Characteristics of $Na_{1+2x}Zr_{2-x}Co_x(PO_4)_3$ ceramics.

| Ceramics | Co, wt.% | Density, g/cm³ | Theoretical Density, g/cm³ | Relative Density, % | Hv, GPa | $K_{IC}$, MPa·m$^{1/2}$ |
|---|---|---|---|---|---|---|
| $Na_{1.2}Zr_{1.9}Co_{0.1}(PO_4)_3$ | 1.20 | 3.151 | 3.187 | 98.87 | 4.7 ± 0.2 | 1.1 ± 0.2 |
| $Na_{1.5}Zr_{1.75}Co_{0.25}(PO_4)_3$ | 2.98 | 3.088 | 3.200 | 96.50 | 5.8 ± 0.4 | 0.7 ± 0.2 |
| $Na_2Zr_{1.5}Co_{0.5}(PO_4)_3$ | 5.93 | 3.006 | 3.222 | 93.30 | 3.0 ± 0.5 | 1.0 ± 0.2 |

Temperature curves of shrinkage L(T) and shrinkage rate S(T) are shown in Figure 9. L(T) temperature curves had a two-stage character. At low heating temperatures, the intensive shrinkage stage characterized by a high shrinkage rate S was observed. At high heating temperatures, the shrinkage rate of the powders was lower than the one at low heating temperatures. Maximum shrinkage values $L_{max}$ for the powders decreased along with an increase in cobalt content in the NZP structure. The $Na_{1.2}Zr_{1.9}Co_{0.1}(PO_4)_3$ specimen was sintered at temperatures ranging from 670 to 800 °C; the maximum shrinkage rate ($S_{max} = 1.1 \cdot 10^{-2}$ mm/s) was reached at 730–750 °C. The $Na_{1.5}Zr_{1.75}Co_{0.25}(PO_4)_3$ specimen started to shrink at a lower temperature: the maximum shrinkage rate $S_{max} = 1.5\ 10^{-2}$ mm/s was reached at 680–705 °C. The second maximum (~0.75·10$^{-2}$ mm/s) in the S(T) temperature curve for the ceramic with x = 0.25 was observed at 780–790 °C, and an essential shift of the intensive powder shrinkage interval towards lower temperatures was observed (Figure 9c). The maximum shrinkage rate $S_{max} = 0.9 \cdot 10^{-2}$ mm/s for the ceramic with x = 0.5 was observed at 635–650 °C. At temperatures exceeding 700 °C, the second maximum in S(T) temperature dependence for $Na_2Zr_{1.5}Co_{0.5}(PO_4)_3$ ceramic was observed.

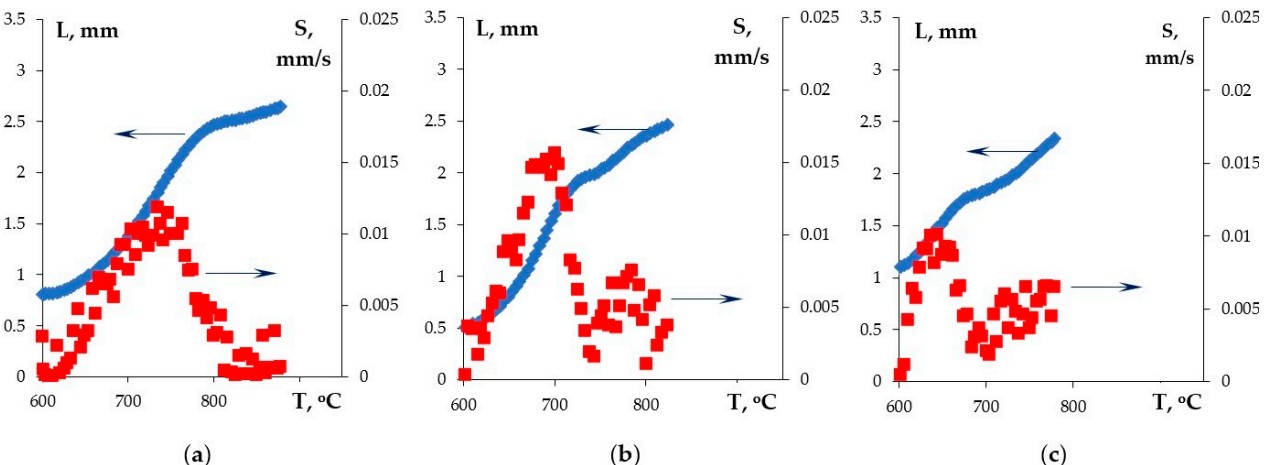

**Figure 9.** Temperature curves of shrinkage value L (blue) and shrinkage rate S (red) for $Na_{1.2}Zr_{1.9}Co_{0.1}(PO_4)_3$ (**a**), $Na_{1.5}Zr_{1.75}Co_{0.25}(PO_4)_3$ (**b**), and $Na_2Zr_{1.5}Co_{0.5}(PO_4)_3$ (**c**) phosphates.

It should be noted that ceramics with increased cobalt contents (x = 0.25, 0.5) displayed a two-stage character of L(T) and S(T) curves at the intensive powder shrinkage stage. In our opinion, this can be linked to agglomerates present in the synthesized powders and, as shown in refs. [2,90], the sintering of the particles "inside" the agglomerates and the baking of the agglomerates to each other work at different rates. Most likely, higher cobalt content leads to an increase in the strength of the agglomerates and, as a consequence, to higher temperatures, at which the regrouping of the submicron particles inside the agglomerates takes place.

A comparison of XRD patterns of initial powders and sintered ceramics (Figure 10) showed that the phase composition of the specimens did not change as a result of sintering. The XRD peaks correspond to the graphite phase, which often forms at the interaction of the ceramics with the graphite die surface (see ref. [105]) and is absent in Figure 10.

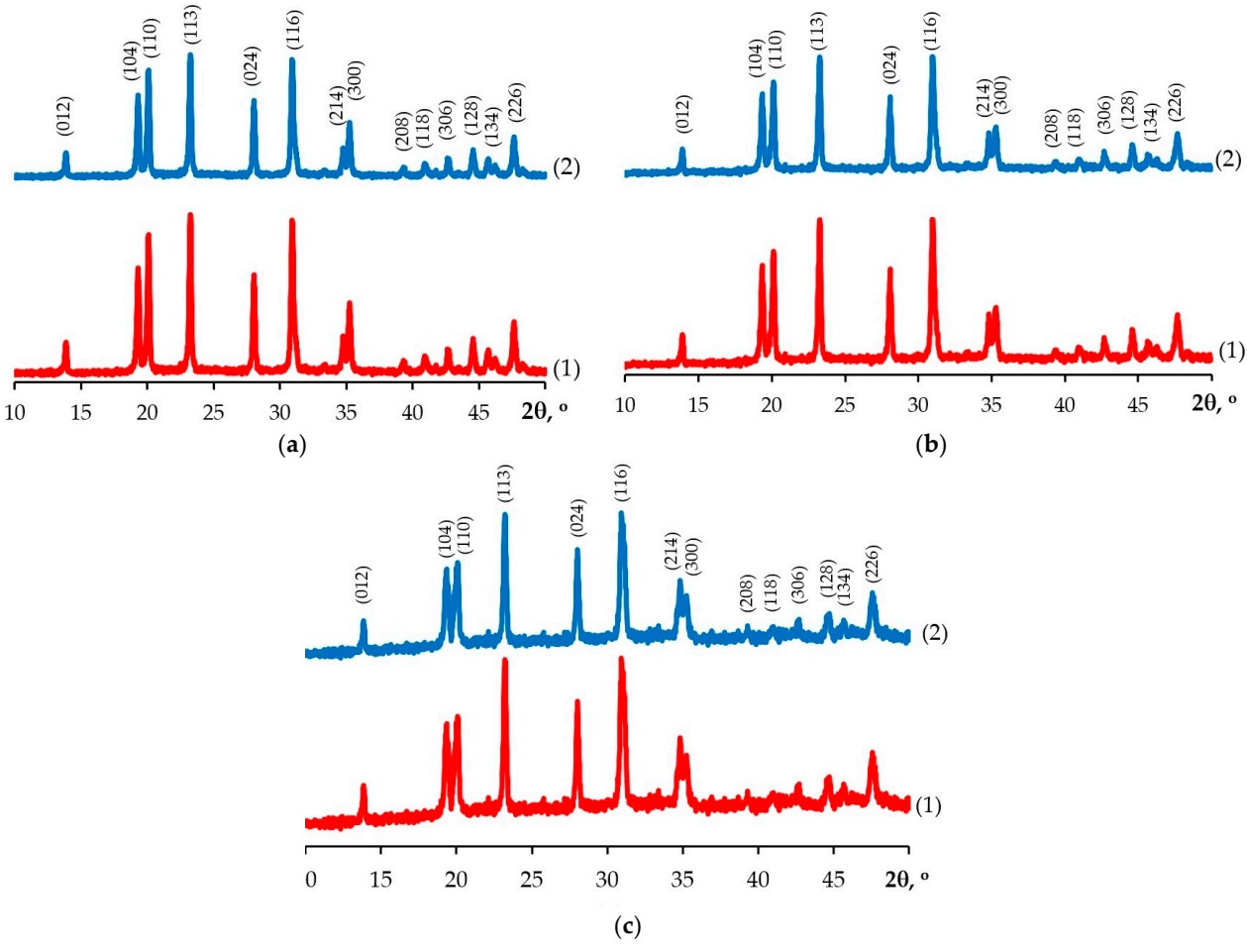

**Figure 10.** XRD data for $Na_{1.2}Zr_{1.9}Co_{0.1}(PO_4)_3$ (**a**), $Na_{1.5}Zr_{1.75}Co_{0.25}(PO_4)_3$ (**b**), and $Na_2Zr_{1.5}Co_{0.5}(PO_4)_3$ (**c**) ceramics: line (1)—powders, line (2)—ceramics.

Analysis of SEM photographs of ceramics (x = 0.1) (Figure 8b) indicates that the resulting ceramics have a fine-grained microstructure, the grain size of which ranges from 0.5 to 1 μm. Abnormal grain growth was observed at higher levels of cobalt in the structure of ceramics. As can be seen in Figure 8f, abnormally large grains of 3–5 μm in size surrounded by submicron-sized grains were observed in the microstructure of ceramics with increased cobalt content (x = 0.5). The results of the semi-quantitative EDS analysis are presented in Table 3. This also confirms that the composition of the ceramics did not change: based on EDS results, the composition of sintered ceramics is close to that of initial powders.

**Table 3.** EDS analysis results of the ceramics $Na_{1+2x}Zr_{2-x}Co_x(PO_4)_3$.

| Ceramics | Chemical Compositions, wt.% | | | | |
|---|---|---|---|---|---|
| | **O** | **Na** | **P** | **Co** | **Zr** |
| $Na_{1.2}Zr_{1.9}Co_{0.1}(PO_4)_3$ | 38.08 | 5.40 | 16.60 | 0.84 | 38.87 |
| $Na_{1.5}Zr_{1.75}Co_{0.25}(PO_4)_3$ | 37.84 | 6.98 | 16.71 | 2.79 | 35.47 |
| $Na_2Zr_{1.5}Co_{0.5}(PO_4)_3$ | 38.30 | 7.92 | 16.65 | 5.30 | 31.58 |

Minimal values of fracture toughness coefficient and microhardness in the ceramics (Table 2) are typical of ceramics based on compounds of this structural type [2,9,21,22]. Note that a decrease in ceramic microhardness from 4.7 GPa down to 3 GPa was observed along with an increase in cobalt content, whereas the minimum fracture toughness coefficient

almost did not change and was ~1 MPa·m$^{1/2}$ (Table 2). In our opinion, a decrease in microhardness accompanied by an increase in cobalt content stems from a decrease in ceramic density as well as microstructure nonuniformities in abnormally large grains (Figure 8f).

Let us analyze the effect of cobalt on the kinetics of rapid sintering in $Na_{1+2x}Zr_{2-x}Co_x(PO_4)_3$ ceramics.

To analyze the kinetics of the rapid sintering of powders at Stage II, let us use the Yang–Cutler model [106,107] describing the initial stage of low-temperature non-isothermal sintering of spherical particles under simultaneous volume and grain boundary diffusion as well as plastic deformation (creep). According to ref. [106], the slope of temperature (T)–relative shrinkage ($\varepsilon$) curve plotted on the $\ln(T\partial\varepsilon/\partial T) - T_m/T$ axes corresponds to the effective sintering activation energy $mQ_{s(1)}$ (in $kT_m$, where k is the Boltzmann constant and $T_m$ is the melting point). The magnitude of $m$ depends on the prevailing sintering mechanism: $m = 1/3$ for grain boundary diffusion, $m = 1/2$ for crystal lattice diffusion, and $m = 1$ for viscous flow (creep). The efficiency of the Yang–Cutler model used for calculating SPS activation energy of mineral-like ceramics was demonstrated in refs. [40,41,90,108–110]. When calculating SPS activation energy, the magnitude of the melting point for all NZP-Co compounds was taken as $T_m = 2073$ K.

Figure 11a presents the $\ln(T\partial\varepsilon/\partial T) - T_m/T$ curves for $Na_{1+2x}Zr_{2-x}Co_x(PO_4)_3$ ceramics (x = 0.1, 0.25, 0.5). These curves have a conventional two-stage character with a maximum (see ref. [106]). The analysis shows an insufficient increase in effective sintering activation energy $mQ_{s(1)}$ with increasing cobalt content from 5.9 $kT_m$ (for x= 0.1) up to 6.3 $kT_m$ (at x = 0.5). The results obtained agree qualitatively with a decrease in maximum shrinkage of the powders observed (Figure 8) and a decrease in the relative density of the sintered ceramics with increased cobalt content (Table 2).

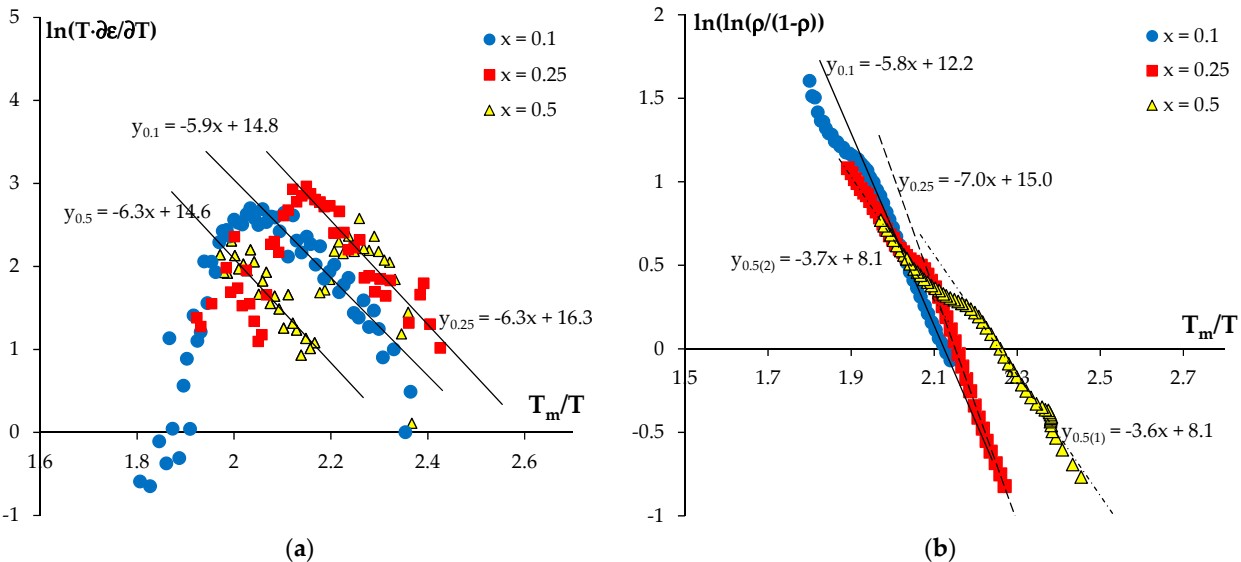

**Figure 11.** $\ln(T\partial\varepsilon/\partial T) - T_m/T$ (**a**) and $\ln(\ln(\alpha\cdot\rho/(1-\rho))) - T_m/T$ (**b**) curves for $Na_{1+2x}Zr_{2-x}Co_x(PO_4)_3$ ceramics with varying cobalt content (x = 0.1, 0.25, 0.5). Determination of SPS activation energy at the stage of intense powder shrinkage.

As can be seen in Figure 11a, the slope of the $\ln(T\partial\varepsilon/\partial T) - T_m/T$ curve in the high-temperature region becomes negative. To estimate the sintering activation energy for the high-temperature stage, other methods of analyzing the shrinkage curve shall be used.

The activation energy at the high-temperature stage of SPS can be estimated using the model of diffusion dissolving of the pores located near grain boundaries in fine-grained ceramic materials [111]. The activation energy at this stage of SPS $Q_{s(2)}$ is determined from the slope of the $\rho(T)/\rho_{th}$ temperature curve in the double logarithmic axes

$\ln(\ln(\alpha\cdot\rho/\rho_{th}/(1 - \rho/\rho_{th})) - T_m/T$, where $\alpha$ is the compaction coefficient of pressing ($\alpha = 0.6$ for $Na_{1+2x}Zr_{2-x}Co_x(PO_4)_3$ powders) (Figure 11b). The mean uncertainty of determining the activation energy $Q_{s(2)}$ is $\pm 0.5$ $kT_m$. The efficiency of this approach used to estimate the SPS activation energies at high heating temperatures was demonstrated earlier in refs. [40,41,90,95–97,108–110].

As one can see in Figure 11b, the temperature curves of shrinkage in the $\ln(\ln(\alpha\cdot\rho/\rho_{th}/(1 - \rho/\rho_{th})) - T_m/T$ axes have a two-stage character. At low temperatures, $\ln(\ln(\alpha\cdot\rho/\rho_{th}/(1 - \rho/\rho_{th})) - T_m/T$ curves can be fitted by straight lines with good precision. At high heating temperatures, the non-monotonous character of $\ln(\ln(\alpha\cdot\rho/\rho_{th}/(1 - \rho/\rho_{th})) - T_m/T$ curves was observed. At low heating temperatures, the magnitudes of activation energy for ceramics with cobalt contents x = 0.1 and x = 0.25 were 5.8 and 7.0 $kT_m$, respectively. So far, the effective values of SPS activation energy for the ceramics with x = 0.1 and x = 0.25 calculated using the Yang–Cutler model ($mQ_{s(1)}$ = 5.8–6.3 $kT_m$) were close to the ones calculated using the model described in ref. [106] ($Q_{s(2)}$ = 5.8–7.0 $kT_m$). Therefore, we can conclude that the magnitude of the *m* coefficient is close to the theoretical one $m \sim 1$. According to refs. [106,107], $m \sim 1$ means that the compaction kinetics of the submicron-grade $Na_{1.2}Zr_{1.9}Co_{0.1}(PO_4)_3$ and $Na_{1.5}Zr_{1.75}Co_{0.25}(PO_4)_3$ powders in SPS are determined by creep intensity. It is quite an unexpected result since usually it is assumed that the SPS kinetic of nano- and fine-grained powders with kosnarite, scheelite, garnet, etc. mineral structures is determined by grain boundary diffusion intensity (see refs. [40,41,90,91,95–97,108–110]). The creep process in SPS is often observed in the cases of alumina [111,112] or tungsten carbide [113,114] at essentially higher heating temperatures. The results obtained suggest that cobalt should be added to the NZP crystal lattice to promote a dislocation motion process.

For $Na_2Zr_{1.5}Co_{0.5}(PO_4)_3$ ceramics, $\ln(\ln(\alpha\cdot\rho/\rho_{th}/(1 - \rho/\rho_{th})) - T_m/T$ curves have a two-stage character expressed clearly (Figure 11b). Note that two linear parts with approximately equal slopes are seen clearly in the $\ln(\ln(\alpha\cdot\rho/\rho_{th}/(\rho/\rho_{th} - 1)) - T_m/T$ curves. The SPS activation energy values calculated using the model described in ref. [110] were abnormally low (3.6–3.7 $kT_m$) and did not match the known data on activation energies of bulk diffusion [115,116] and grain boundary diffusion [115,117] in the ceramics. So far, we cannot explain such low SPS activation energy values in $Na_2Zr_{1.5}Co_{0.5}(PO_4)_3$ ceramics. Note that XRD phase analysis and TEM did not find any anomalies in the structure, microstructure, or phase composition of the powder synthesized.

## 4. Conclusions

New $Na_{1+2x}Zr_{2-x}Co_x(PO_4)_3$ phosphates (x from 0 to 0.5) with $NaZr_2(PO_4)_3$ structure (NZP, NASICON) were obtained and described. According to the data on high-temperature X-ray diffraction for 25–700 °C, a linear dependence of the crystal lattice parameters *a* and *c* on temperature was established: *a* values decreased and *c* values increased after heating. It was found that adding cobalt led to an increase in the average coefficient of thermal expansion (CTE). It has been shown that $Na_{1+2x}Zr_{2-x}Co_x(PO_4)_3$ phosphates (x = 0.1, 0.2, 0.3, 0.4, 0.5) belong to the class of materials that expand moderately after heating ($2\cdot10^{-6} \leq \alpha_{av} \leq 8\cdot10^{-6}$ °C$^{-1}$).

It becomes possible to create materials with specified parameters of thermal expansion using the identified regularities regarding the substitution effect in the cationic part of the framework on the thermal expansion of $NaZr_2(PO_4)_3$ compounds.

Taking $Na_{1+2x}Zr_{2-x}Co_x(PO_4)_3$ compositions (x from 0.1 to 0.5) as an example, the possibility of obtaining a ceramic specimen using SPS with high relative density (up to 98.87%) and high strength characteristics was demonstrated. It was shown that the density of the sintered specimens decreases, and the SPS activation energy calculated within the framework of the Yang–Cutler model increases with increasing cobalt content. The intensity of rapid sintering of submicron-grade $Na_{1+2x}Zr_{2-x}Co_x(PO_4)_3$ powders was found to be limited by creep processes and diffusion in crystal lattice under uniaxial stress (70 MPa).

**Author Contributions:** Conceptualization, A.I.O.; formal analysis, A.I.O., A.A.A., V.N.C., and S.G.-G.; investigation, A.A.A., D.O.S., M.S.B., S.A.K., A.A.M., A.A.P., G.V.S., and N.Y.T.; methodology, A.I.O., A.A.A., and A.V.N.; resources, V.N.C., A.V.N., and S.G.-G.; data curation, A.I.O., A.A.A., and A.V.N.; writing—original draft preparation, A.A.A., A.I.O., and A.V.N.; writing—review and editing, A.I.O., A.V.N., and V.N.C.; visualization, A.A.A. and A.V.N.; supervision, A.I.O. and V.N.C.; project adminis­tration, A.I.O.; funding acquisition, A.I.O. All authors have read and agreed to the published version of the manuscript.

**Funding:** This research was funded by the Russian Science Foundation, grant number 21-13-00308. TEM investigations of the microstructure were carried out using the equipment of the Center Collec­tive Use "Materials Science and Metallurgy" (National University of Science and Technology "MISIS") with the financial support of the Ministry of Science and Higher Education of the Russian Federation (Grant number 075-15-2021-696).

**Institutional Review Board Statement:** Not applicable.

**Informed Consent Statement:** Not applicable.

**Data Availability Statement:** Not applicable.

**Conflicts of Interest:** The authors declare no conflict of interest.

## Appendix A

*Features of the Crystal Structure of NZP Compound*

At present, there are many reviews in which the structures of NZP compounds are described well enough (see, for example, refs. [1,3,6–8,25–28,30,31,35,41–55,58,60,61,65–68, 71,72,74,118]). We will not concentrate on this issue in detail here. Below, you can find only key data which is necessary for the analysis of the results obtained.

$NaZr_2(PO_4)_3$ compounds crystallize in the trigonal syngony, the rhombohedral sym­metry space group of $R\bar{3}c$. The unit cell parameters of the NZP compound are: $a = b = 8.8045(2)$ Å, $c = 22.7585(9)$ Å, unit cell volume V = 1530.5 Å$^3$, X-ray density 3.22 g/cm$^3$, and number of formula units in the unit cell N = 6. The NZP compound has a frame­work structure in which zirconium, along with phosphorus, plays the anion-forming role (Figure A1).

$NaZr_2(PO_4)_3$ phosphate and its structural analogs are described with the general crystal chemical formula $(M1)^{VI}(M2)^{VIII}_3(L^{VI}_2(PO_4)_3]$, where M1(Na1) and M2(Na2) are the positions in the framework planes, L stands for framework positions occupied by zirconium, and the Roman digits are the co-ordination surroundings (co-ordination number, CN) in these positions.

The framework base is constituted by the structural units $[Zr_2(PO_4)_3]$ consisting of two zirconium octahedrons $ZrO_6$ ($d_{Zr-O} = 2.048–2.084$ Å) and three phosphate octahedrons $PO_4$ ($d_{P-O} = 1.516–1.546$ Å) connected by common oxygen atoms (Figure A1b). $Zr_2(PO_4)_3$ connected to each other form a rigid three-dimensional anion framework consisting of $[O_3(Zr)O_3(PO_4)_3]_\infty$ bands along the crystallographic axis $c$ (Figure A1a).

The electrical neutrality of the structure is provided by the population of the framework voids M with Na cations (Figure A1c). The M1(Na1) position is located in the trigonal antiprism (CN = 6) formed by the triangular facets of two adjacent octahedron $ZrO_6$, in the vertices of which the oxygen atoms are located. There are four void 1(M1):3(M2) per each formula unit available for a population with metal ions. Furthermore, M3 positions are distinguished in the NZP structure, which is co-ordinated by the triangular prism formed by the side facets of the zirconium octahedrons belonging to the same $Zr_2(PO_4)_3$ blocks. However, these ones are too small to accommodate even small cations.

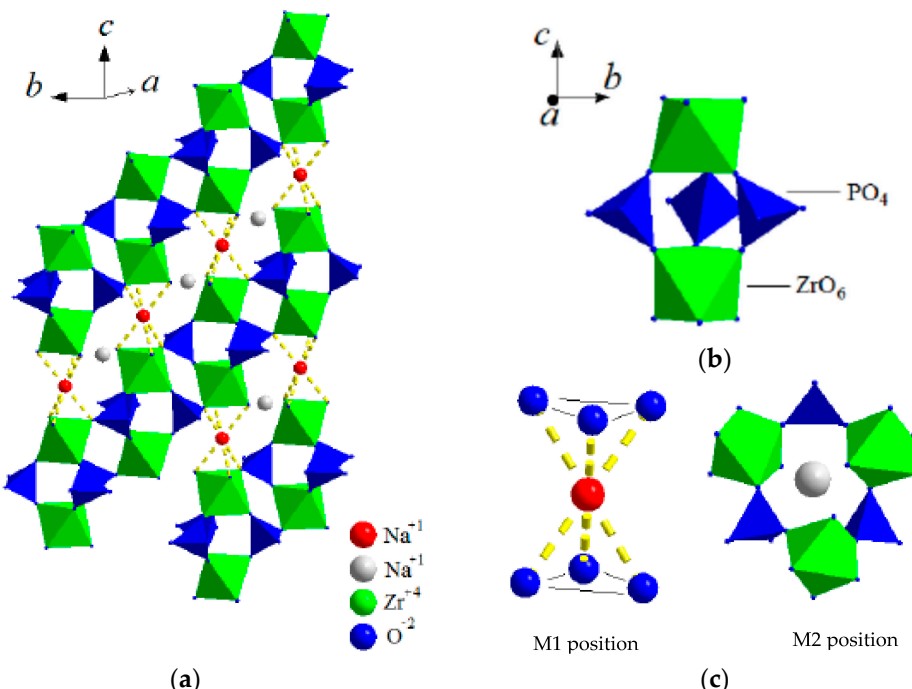

**Figure A1.** Fragment of the NaZr$_2$(PO$_4$)$_3$ structure (**a**), structural unit of the Zr$_2$(PO$_4$)$_3$ framework (**b**), and co-ordination polyhedra of framework cavities (**c**) [43].

The difference in the co-ordination surroundings of cations in void positions M1 (trigonal antiprism) and M2 (octagon) determines their energetic non-equivalence. According to the refined structural model of NaZr$_2$(PO$_4$)$_3$ sodium–zirconium phosphate, the population of M2 positions at room temperature can take place, but the population degree is insufficient and increases up to 10% at 993 K [119]. It probably originates from the presence of an energy barrier preventing the free diffusion of atoms from one void position to another.

It is worth noting that the compounds with the NZP structure are extremely versatile because of the possibility of incorporating various cations into various crystallographic structural positions [67,120].

The heterovalent substitution of cations in the L positions leads to changes in the negative charge of the framework from 0 up to 4 [121]. It is compensated by means of various populations of M positions. As mentioned above, framework void positions M1 and M2 can be populated completely or partly as well as remain vacant. M positions are populated preferentially by low-charged and relatively large cations, whereas the framework is constituted by multi-charged cations with small sizes and the oxidation degrees 5+, 4+, 3+, or 2+ and anions XO$_4$ where X is Si, P, V, As, S, or Mo.

There are some deviations from this crystal chemical principle [68,122–126] but, as a rule, these are featured by a decrease in the thermal stability range of the structure, the manifestation of polymorphism, and problems in synthesis (the changes in the synthesis kinetics and reduction of the synthesis rate).

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
