# Peer review of "Spark Plasma Sintering of Ceramics Based on Solid Solutions of Na1+2xZr2−xCox(PO4)3 Phosphates: Thermal Expansion and Mechanical Properties Research"

_ceramics, doi:10.3390/ceramics6010017_

Round 1

Reviewer 1 Report

Aleksandrow et al. report on the manufacturing of a solid solution of Na1+2xZr2-xCox(PO4)3 by the SPS method. The products are characterized mainly by X-ray powder diffraction and SEM. While the scientific work seems adequately performed, the submission must be rejected for the very sloppy presentation. In the following, I will only comment on the abstract in detail and leave the careful revision of the text to the ten authors.

A new submission might be appropriate for publication.

Abstract:

L17: 04

L17: remove “containing cobalt”

L18: The solid solution Na1+2xZr2-xCox(PO4)3 was obtained by ….

Annealing of… ? // Reaction of..? //At which temperature? There is a temperature dependence for solubilities.

L19: Which silicates? Did you also substitute the phosphate group?

L23: “were obtained”. Provide the density value.

L24: “intensity of crystal lattice diffusion”    …is controlled by atomic diffusion

Figures

Fig. 1 The labels are too small (in some other figures as well). Either “1,2,3” or “a,b,c”

Fig. 3 Provide a calculated line diagram. The weak reflections at around 40° belong to the phase?

Fig.8 / Fig.9: ??? Seriously?

Fig 10 Please use appropriate symbols.

General remarks to the text (not complete):

Free internet tools can find many mistakes concerning language.

To my knowledge, Kosnarite is the K-containing compound. With Na, it is “Kosnrate type”. Please check.

The introduction should refer to some other studies of solid solutions of NZP. NASICON (please mention) is a well-known ion conductor. Explain why you expect high stability by your substitution.

Your preparation method is based on the thermal decomposition of the educts. Find a more appropriate description than the “solid-phase method”.

Author Response

The authors thank the Reviewer for considering the paper. We agree with most of the remarks made. Below please find our response to the Reviewer comments.

Reviewer #1

Abstract

Comment 1. L17: 04

Comment response: Corrected

Comment 2. L17: remove “containing cobalt”

Comment response: Corrected

Comment 3. L18: The solid solution Na1+2xZr2-xCox(PO4)3 was obtained by …

Comment response: Corrected

Comment 4. Annealing of… ? // Reaction of..? //At which temperature? There is a temperature dependence for solubilities.

Comment response: The Abstract contains additional info about the process of synthesis.

Comment 5. L19: Which silicates? Did you also substitute the phosphate group?

Comment response: Corrected

Comment 6. L23: “were obtained”. Provide the density value.

Comment response: Corrected

Comment 7. L24: “intensity of crystal lattice diffusion”    …is controlled by atomic diffusion

Comment response: Corrected

Figures

Comment 8. Fig. 1 The labels are too small (in some other figures as well). Either “1,2,3” or “a,b,c”

Comment response: Corrected

Comment 9. Fig. 3 Provide a calculated line diagram. The weak reflections at around 40° belong to the phase?

Comment response: Corrected

Comment 10. Fig.8 / Fig.9: ??? Seriously?

Comment response: Corrected

Comment 11. Fig 10 Please use appropriate symbols.

Comment response: Figure 10 now has appropriate symbols.

General remarks

Comment 12. To my knowledge, Kosnarite is the K-containing compound. With Na, it is “Kosnarite type”. Please check.

Comment response: Corrected

Comment 13. The introduction should refer to some other studies of solid solutions of NZP. NASICON (please mention) is a well-known ion conductor. Explain why you expect high stability by your substitution.

Comment response: Corrected

Comment 14. Your preparation method is based on the thermal decomposition of the educts. Find a more appropriate description than the “solid-phase method”.

Comment response: We do not agree with the remark. The term "solid-phase method" is widely used in many research works.

Reviewer 2 Report

Comments and Suggestions for Authors

 The manuscript „Spark Plasma Sintering of ceramics based on solid solutions of Na1+2xZr2-xCox(PO4)3 phosphates. Thermal expansion research” by A.A. Aleksandrov, A.I. Orlova, D.O. Savinykh, M.S. Boldin, S.A. Khainakov, A.A. Murashov, A.A. Popov, G.V. Shcherbak, S. Garcia-Granda, A.V. Nokhrin, V.N. Chuvil’deev, and N.Yu. Tabachkova  reports synthesis and characterization of new ceramic materials  based on solid solution of Na1+2xZr2-xCox(PO4)3 phosphates for 0>x>0.5. New phases obtained by the solid-phase method crystallize in the structure of the NaZr2(PO4)3 type (trigonal system, space group R3̅c). As part of this work was examined thermal expansion of the new phases by high-temperature X-ray diffraction in the temperature range from 25 to 700°C. The parameters of thermal expansion are calculated and their dependence on the composition of samples were analyzed. High-density ceramics Na1+2xZr2-xCox(PO4)3  obtained by the Spark Plasma Sintering (SPS). It is shown that the kinetics of the SPS process is determined by the intensity of crystal lattice diffusion. It is found that the density of ceramics decreases and the activation energy of SPS ceramics increases  as the cobalt content increases, and anomalous grain growth in ceramics is observed.

Some sugestions in your work will make your manuscript more suitable to be published in Ceramics:

1.   The abbreviation NZP given in the keywords should be expanded.

2.   Figure 2 does not provide relevant information as it cannot be determined whether they are different diffraction patterns. There is no apparent shift in the diffraction lines. It seems advisable to complete the drawing and provide the value of 2θ for each diffraction line belonging to different samples of a solid solution.

3.   Please explain how, on the basis of the EDS results of the analyzed samples, presented in Table 3, the authors claim that the composition of the samples does not change. In the work, 7 samples of a solid solution Na1+2xZr2-xCox(PO4)3 for x =0; 0.1; 0.2; 0.25; 0.3; 0.4; 0.5 were obtained, it would be beneficial that the results of EDS tests for all these samples were included by the authors in Table 3.

Overall, after completing these data, the paper should be accepted.

Author Response

The authors thank the Reviewer for considering the paper. We agree with most of the remarks made. Below please find our response to the Reviewer comments.

Reviewer #2

Comment 1. The abbreviation NZP given in the keywords should be expanded.

Comment response: Corrected. The NZP abbreviation is expanded in the Abstract and in the first line of the Introduction.

Comment 2. Figure 2 does not provide relevant information as it cannot be determined whether they are different diffraction patterns. There is no apparent shift in the diffraction lines. It seems advisable to complete the drawing and provide the value of 2θ for each diffraction line belonging to different samples of a solid solution.

Comment response: The figure has been corrected. The text of the article contains a link to NaZr2(PO4)3 compound card in the PDF database, which contains all the relevant information about diffraction angles for each of the XRD maxima.

Comment 3. Please explain how, on the basis of the EDS results of the analyzed samples, presented in Table 3, the authors claim that the composition of the samples does not change. In the work, 7 samples of a solid solution Na1+2xZr2-xCox(PO4)3 for x =0; 0.1; 0.2; 0.25; 0.3; 0.4; 0.5 were obtained, it would be beneficial that the results of EDS tests for all these samples were included by the authors in Table 3.

Comment response: Corrected. Ceramic samples were sintered using powders with x = 0.1, 0.25, and 0.5. Ceramics with intermediate concentrations of cobalt were not sintered.

Reviewer 3 Report

The manuscript „Spark Plasma Sintering of ceramics based on solid solutions of 2 Na1+2xZr2-xCox(PO4)3 phosphates. Thermal expansion research” represents a valuable contribution to the sintering of Co containing Na-Zr-Phosphates and should be published after further revision. The following comments should be considered:

1)        The heading should not be interrupted with a punctation mark. I suggest a change to: "Spark Plasma Sintering of ceramics based on solid solutions of 2 Na1+2xZr2-xCox(PO4)3 phosphates: Thermal and mechanical characteristics”.

2)               Page 1, line 18: What does “on the basis of solid solutions” means? Do you mean “solid educts” or “oxides” or "salts"?

3)               Page 1, line 19: Do you mean phophates or silicates? I do not understand, why silicates are obtained.

4)               Page 1, line 25-27: “are carried out”, “was shown”, “was found”, “were observed” (grammar). Please stay in the same tense when you are explaining the experimental work. Present progessive is better than past progressive in the abstract.

5)               Page 2, line 49: “the phases in which have” (grammar) could be shorten to “which have”.

6)               Page 4, line 168/169: Later, you compare the relative density of the samples. Please give the theoretical value of the density (100%), to understand the calculation of the relative density.

7)               Page 4, line 174-176: Did you polish the surface before measuring the hardness? Please describe the procedure.

8)               Page 5, line 197: In “и Na2Zr1.5Co0.5(PO4)3” The Cyrillic letter seems to be redundant.

9)               Figure 3: Intensity is given in “arb. unit”. Please label the axis with the unit.

10)            Figure 4: Please use English punctation on the axis. Please define again in the caption, what X is (comparable to Figure 3).

11)            Figure 5/6: Please use English punctation on the axis.

12)            Page 8, line 224: “were cannot” means “we cannot”. Please correct.

13)              Page 8, line 236/237: The whole sentence is not clearly understandable.

14)              Figure 7, caption: “of the pressure applied” should be changed to “the applied pressure”.

15)              Table 2: Unit of the density has to be “g/cm³”.

16)              Page 8, line 245/246: Which theoretical density was used for calculation of the relative density? Was the theoretical density adjusted to change of the Co content?

17)              Figure 8, Caption: Better to start with “SEM images of …” and delete “SEM” at the end.

18)              Page 10, line 273: The second part of the sentence is not clearly understandable. “,which the regrouping …” could be substitutes by “whereas the regrouping …”.

19)              Figure 10: Caption does not fit to the XRD patterns.

20)              Page 11, line 284: Please improve the expression of the subordinate clause (“,the grain size of which …”).

21)              Page 11, line 300 and 302: Sentences in the active voice should be written in the passive voice.

22)              Page 12, line 326: Expression “It means that one has to use” should be improved to clarify the meaning of the sentence.

23)              Page 12, line 345: “It allows concluding the” should be improved by using “In conclusion, the …”.

24)              Page 13, line 370, 371, 374 and 379: Please avoid “It …” sentences.

In general, the experimental work and the presentation of results as well as the discussion are well done. I recommend using frame lines for the figures and to take care on the punctuation. The scientific content is substantial and a deep insight into the literature is given. I recommend minor revision with focus on grammar and expression.

Author Response

The authors thank the Reviewer for considering the paper. We agree with most of the remarks made. Below please find our response to the Reviewer comments.

Reviewer #3

Comment 1. The heading should not be interrupted with a punctation mark. I suggest a change to: "Spark Plasma Sintering of ceramics based on solid solutions of 2Na1+2xZr2-xCox(PO4)3 phosphates: Thermal and mechanical characteristics”.

Comment response: Corrected

Comment 2. Page 1, line 18: What does “on the basis of solid solutions” means? Do you mean “solid educts” or “oxides” or "salts"?

Comment response: Corrected

Comment 3. Page 1, line 19: Do you mean phophates or silicates? I do not understand, why silicates are obtained.

Comment response: Corrected

Comment 4. Page 1, line 25-27: “are carried out”, “was shown”, “was found”, “were observed” (grammar). Please stay in the same tense when you are explaining the experimental work. Present progessive is better than past progressive in the abstract.

Comment response: Corrected

Comment 5. Page 2, line 49: “the phases in which have” (grammar) could be shorten to “which have”.

Comment response: Corrected

Comment 6. Page 4, line 168/169: Later, you compare the relative density of the samples. Please give the theoretical value of the density (100%), to understand the calculation of the relative density.

Comment response: Corrected (see Table 2).

Comment 7. Page 4, line 174-176: Did you polish the surface before measuring the hardness? Please describe the procedure.

Comment response: Corrected

Comment 8. Page 5, line 197: In “и Na2Zr1.5Co0.5(PO4)3” The Cyrillic letter seems to be redundant.

Comment response: Corrected

Comment 9. Figure 3: Intensity is given in “arb. unit”. Please label the axis with the unit.

Comment response: Corrected

Comment 10. Figure 4: Please use English punctation on the axis. Please define again in the caption, what X is (comparable to Figure 3).

Comment response: Corrected

Comment 11. Figure 5/6: Please use English punctation on the axis.

Comment response: Corrected

Comment 12. Page 8, line 224: “were cannot” means “we cannot”. Please correct.

Comment response: Corrected

Comment 13. Page 8, line 236/237: The whole sentence is not clearly understandable.

Comment response: Corrected

Comment 14. Figure 7, caption: “of the pressure applied” should be changed to “the applied pressure”.

Comment response: Corrected

Comment 15. Table 2: Unit of the density has to be “g/cm³”.

Comment response: Corrected

Comment 16. Page 8, line 245/246: Which theoretical density was used for calculation of the relative density? Was the theoretical density adjusted to change of the Co content?

Comment response: Corrected (see Table 2)

Comment 17. Figure 8, Caption: Better to start with “SEM images of …” and delete “SEM” at the end.

Comment response: Corrected

Comment 18. Page 10, line 273: The second part of the sentence is not clearly understandable. “,which the regrouping …” could be substitutes by “whereas the regrouping …”.

Comment response: Corrected

Comment 19.   Figure 10: Caption does not fit to the XRD patterns.

Comment response: Corrected

Comment 20. Page 11, line 284: Please improve the expression of the subordinate clause (“,the grain size of which …”).

Comment response: Corrected

Comment 21. Page 11, line 300 and 302: Sentences in the active voice should be written in the passive voice.

Comment response: Corrected

Comment 22. Page 12, line 326: Expression “It means that one has to use” should be improved to clarify the meaning of the sentence.

Comment response: Corrected

Comment 23. Page 12, line 345: “It allows concluding the” should be improved by using “In conclusion, the …”.

Comment response: Corrected

Comment 24. Page 13, line 370, 371, 374 and 379: Please avoid “It …” sentences.

Comment response: Partially corrected

Reviewer 4 Report

This is a good report to the properties of doped NZP with a good review of the development of the research.  I have the following questions and suggestions.

1. Fig 2e, f are mentioned in line 192 but are not included in the submission.

2. Lattice constants are reported in Figure 4.  Are those measured at 25 C?

3. It will help the readers to explain how the average CTE (αav) are calculated.

4. Two symbols of CTE are used in the work αav and αcp.

5. The word "were" in line 224 should be "we".

6. Two sets of data are shown in Figure 9.  The authors should indicate which one (blue line) is L and which one (red dots) is S.

7. The authors can convert the wt% in Table 3 to stoichiometric numbers to show the ratio among metals.

8. "Tm" is used a lot in this work but without any definition or explanation.

9. The authors spend a lot of paragraph to discuss the estimate of activation energy but what is the final estimate?

Author Response

The authors thank the Reviewer for considering the paper. We agree with most of the remarks made. Below please find our response to the Reviewer comments.

Reviewer #4

Comment 1. Fig 2e, f are mentioned in line 192 but are not included in the submission.

Comment response: Corrected

Comment 2. Lattice constants are reported in Figure 4.  Are those measured at 25 C?

Comment response: Figure caption corrected.

Comment 3. It will help the readers to explain how the average CTE (αav) are calculated.

Comment response: Corrected. The description of the calculation procedure has been added to the Materials and Methods section.

Comment 4. Two symbols of CTE are used in the work αav and αcp.

Comment response: Corrected

Comment 5. The word "were" in line 224 should be "we".

Comment response:

Comment 6. Two sets of data are shown in Figure 9.  The authors should indicate which one (blue line) is L and which one (red dots) is S.

Comment response: Corrected

Comment 7. The authors can convert the wt% in Table 3 to stoichiometric numbers to show the ratio among metals.

Comment response: Corrected

Comment 8. "Tm" is used a lot in this work but without any definition or explanation.

Comment response: Corrected. Tm – melting point.

Comment 9. The authors spend a lot of paragraph to discuss the estimate of activation energy but what is the final estimate?

Comment response: Corrected

Round 2

Reviewer 1 Report

Thank you for revising the paper, which went from one extreme to another. I like the text very much, and I look forward to its publication.

Line 206-208: I prefer X-ray "reflections" instead of peaks

Line 295 check the sentence

Line 384 Line break 10-6